# The da Vinci Single-Port Robotic Platform in General Surgery: A Scoping Review of Current Applications and Outcomes

**DOI:** 10.3390/jcm14228212

**Published:** 2025-11-19

**Authors:** Silvio Caringi, Antonella Delvecchio, Annachiara Casella, Cataldo De Palma, Valentina Ferraro, Rosalinda Filippo, Matteo Stasi, Nunzio Tralli, Tommaso Maria Manzia, Michele Tedeschi, Riccardo Memeo

**Affiliations:** 1Unit of Hepato-Biliary and Pancreatic Surgery, “F. Miulli” General Hospital, Acquaviva delle Fonti, 70021 Bari, Italy; a.delvecchio@miulli.it (A.D.); a.casella@miulli.it (A.C.); v.ferraro@miulli.it (V.F.); r.filippo@miulli.it (R.F.); matteo.stasi@miulli.it (M.S.); n.tralli@miulli.it (N.T.); m.tedeschi@miulli.it (M.T.); r.memeo@miulli.it (R.M.); 2Department of Surgery, Università Degli Studi Roma “Tor Vergata”, Via Montpellier 1, 00133 Rome, Italy; 3Department of Pancreatic Surgery, IRCCS Humanitas Research Hospital, Rozzano, 20089 Milan, Italy; cataldo.depalma@st.hunimed.eu; 4Transplant and HPB Unit, Department of Surgery Sciences University of Rome Tor Vergata, 00133 Rome, Italy; manzia@med.uniroma2.it; 5Department of Medicine and Surgery, LUM University, Casamassima, 70010 Bari, Italy

**Keywords:** da Vinci Single-Port (SP) system, robot-assisted general surgery, minimally invasive surgery

## Abstract

**Introduction**: The da Vinci Single-Port (SP) robotic system represents a newer minimally invasive surgical development with greater articulation and reduced surgical footprint through the use of a single incision. While originally applied in urology and otolaryngology, its application in general surgery is on the rise. This review aims to delineate the current applications, outcomes, and limitations of the SP platform in general surgical procedures. **Methods**: A descriptive literature search of PubMed, Scopus, and Embase databases was conducted to identify relevant peer-reviewed studies up to September 2025. The included studies reported SP robotic surgeries in various fields of general surgery. **Results**: A growing body of literature was found that reports the safety and feasibility of SP robotic surgery within general surgery. Advantages reported include improved cosmesis, decreased postoperative pain, and shorter recovery time. The present evidence is largely made up of small case series and initial feasibility studies. Technical drawbacks, such as crowding of instruments and a learning curve, remain issues. **Conclusions**: The da Vinci SP system shows promising potential for application in general surgery, particularly for certain procedures. Additional prospective studies and larger case series need to outline its long-term results, cost-effectiveness, and optimal indications.

## 1. Introduction

Robot-assisted surgery has become an accepted minimally invasive method in every surgical specialty, spurred by technological advancements in visualization, instrument articulation, and ergonomics relative to conventional laparoscopy. The use of multi-port da Vinci systems facilitated the evolving steps toward diminishing incisional number and intra-abdominal trauma to the extent of single-port (SP) robotic systems designed to permit an actual single-site entry with preservation of wristed instruments and 3-D visualization. The da Vinci SP platform (Intuitive Surgical) has been of clinical interest in general surgical settings because of the potential for twin advantages from minimally invasive access and single-incision benefits: improved cosmesis, fewer port-site complications, and potential for new access routes (e.g., trans-umbilical, extraperitoneal). This is a review of the SP platform design, its broad surgical uses, evidence regarding outcomes and safety, adoption realities, economic and training implications, and research.

## 2. Device Features and Technical Concepts

The da Vinci SP system was planned to offer multi-instrument articulation via a single cannula. The SP has a multi-channel flexible camera and three articulating instruments, which are placed inside the abdominal cavity and achieve internal triangulation outside the single incision. This is contrary to the single-port laparoscopy, which typically relies on rigid straight instruments through a shared access with external crowding and limited triangulation. The SP platform preserves wristed end-effectors and high-definition stereoscopic visualization and minimizes external instrument collisions and port-site incisions [1,2].

The key components are an articular cannula, an articulating 3D endoscope, three multi-jointed instruments spaced to fan out within the cavity, and an external arm for docking. Internal triangulation and flexible optics of the platform allow access to skinny or confined areas (e.g., deep pelvis, retroperitoneum) where otherwise external instrument geometry would restrict motion. Technical literature and initial device descriptions from Intuitive Surgical (Sunnyvale, CA, United States) indicate alternative access methods (narrow access, extraction-site procedures, trans-anal/trans-oral routes) that SP architecture facilitates [2,3].

In practice, SP surgery involves re-planning of ports, a particular docking workflow, and instrument choice consideration since the SP system conventionally used a smaller number of instruments (energy devices, graspers, scissors) compared with established multi-port systems. Stapling device availability and other adjuncts have grown in recent years, which significantly affects the number of general surgical procedures achievable [3,4].

## 3. Methods

A scoping review of the literature was conducted to gather the most up-to-date uses, outcomes, and limitations of the da Vinci Single-Port (SP) robot system in general surgery.

Figure 1 shows a PRISMA-lite flowchart illustrating the selection of studies included in this scoping review.

English-language publications up to September 2025 in the PubMed, Scopus, and Embase databases were searched for the following search terms and combinations:“da Vinci SP” or “Single-Port robot” or “Single-site robotic”;“general surgery” or “abdominal surgery” or “cholecystectomy” or “colorectal” or “gastrectomy” or “pancreas” or “liver” or “adrenalectomy” or “hernia”.

Reference lists of the included reviews and papers were hand-searched to find other studies.

Studies were included if:Reported clinical use of the da Vinci SP system in any field of general surgery (e.g., hepatobiliary, colorectal, gastric, foregut, hernia).Peri-operative or short-term reported outcomes (operative time, blood loss, conversion, complications, or length of stay).Were peer-reviewed clinical publications, case series, cohort studies, or comparative studies.

Exclusion criteria included:Non-clinical trials (cadaveric or animal models).Abstracts, letters, or conference proceedings with non-extractable clinical data.Studies on other robot systems (e.g., multi-port Xi/Si) without an identified SP subgroup.

Two reviewers independently screened titles and abstracts, then full-text review to confirm eligibility. Data retrieved included the type of procedure, number of cases, operative statistics, complications, conversion rate, and significant clinical outcomes.

Results were grouped by anatomic field (pancreas, liver, gallbladder, hernia, colorectal, gastric, adrenal, others).

Because of heterogeneity and descriptive aims, no quantitative meta-analysis was performed. Instead, a narrative synthesis was developed to highlight procedural safety, feasibility, and describe advantages or limitations across studies.

The overall quality of the evidence was quantitatively measured descriptively by study design (retrospective vs. prospective), number of samples, and risk of bias (case-series nature, institutional experience, selection bias).

No organized scoring system (e.g., ROBINS-I or GRADE) was applied due to the limited number and non-comparative nature of studies available.

## 4. Indications and Procedural Applications in General Surgery

### 4.1. Liver and Pancreas

The pancreas has been a natural initial target for SP experimentation, particularly for tumor enucleation and distal pancreatectomy, less technically challenging operations than pancreaticoduodenectomy and not involving extensive vascular reconstruction.

One of the largest initial experiences is a consecutive series of twenty-three patients in which eleven distal pancreatectomies, eleven enucleations, and one pancreaticoduodenectomy were carried out between December 2021 and February 2022 with the SP robot by surgeons. The average operating time was around 156 min, and the median blood loss was around 40 mL. Notably, no patient required conversion to open surgery, and there was no requirement for transfusions. Postoperative morbidity was minimal and consisted of isolated cases of grade B pancreatic fistula or abdominal infection treated conservatively. The median length of stay was about four days, and the tumors were small, with an average diameter of just under three centimeters [1].

Other researchers have documented feasibility by presenting their own case reports. Choi et al., for instance, presented three SP + one-port distal pancreatectomies with splenectomy in individuals whose mean age was around seventy years and body-mass index was almost twenty-eight; all three procedures were performed without conversion, with less than 500 mL of blood loss and uneventful recovery [5].

The isolated series go on further. A single uneventful SP robotic pancreatectomy for serous cystadenoma was performed in just fifty-five minutes with low blood loss and discharge on day 3 [6]. Cumulatively, the early series suggest that, in well-selected benign or low-grade lesions, SP robotic pancreatectomy can be as safe as standard multi-port techniques.

A variety of themes are repeated in these accounts. Most of all, the selection of patients has been wise: the lesions are small (one to three centimeters) and are located in the pancreatic body or tail, where it is simplest to dissect. Several surgeons have used an “SP + one-port” technique, inserting a small ancillary trocar to utilize for staplers or retraction—an acknowledgment that the SP instrument set available can be limiting when handling splenic vessels or the transection of the pancreas [5].

Risk of post-operative pancreatic fistula, the greatest concern with any distal pancreatectomy, has not been overestimated; in the 23-case series, for example, only a single grade B fistula was encountered and managed non-operatively [1]. Operative times are extremely variable, both reflecting the technical difficulty of some operations (especially spleen-preserving ones) and the inevitable learning curve [7].

If the pancreas is a “point of entry” for SP robotics by nature, then the liver presents even greater challenges: large parenchymal transections, management of inflow and outflow, and the constant threat of massive bleeding. Because of these factors, SP liver surgery has been available to date almost exclusively in the form of limited resections and for properly selected patients.

A classic example is a simple SP left lateral sectionectomy performed in 2022 in a patient with intrahepatic duct stones. Docking time was just eight minutes; operating time was around 135 min; estimated blood loss was 50 mL. Pain scores were extremely mild—about one out of ten by the second postoperative day—and the patient was safely discharged home on day 5 [8].

Even in oncology, the platform has been achieved. An 18-year-old woman with recurrent fibrolamellar hepatocellular carcinoma had SP robotic resection of a segment 4 lesion and resection of peritoneal and cardiophrenic lymph-node metastases. The procedure was approximately five hours long, but blood loss was only 37 mL, margins were negative, and she was discharged on day 8 without complications [9].

There is even a case series of robotic caudate lobectomy with the SP system in a three-centimeter Spiegel lobe lesion; once again, the outcome was good, with discharge on the sixth postoperative day [10]. As with the pancreas, these operations in the liver share some common features. Peripheral, left-sided, or caudate lesions have been selected by surgeons, and major hepatectomies have been avoided. The SP is most often placed to optimize the angle of approach, and meticulous haemostasis is essential. At present, the insufficient availability of SP-specific staplers and parenchymal transection equipment makes it rare for operations involving rapid transection of large quantities of parenchyma to be undertaken.

But the quoted oncologic quality has been reassuring; there was a negative margin on resection for the fibrolamellar carcinoma [9], and blood loss throughout the operations in all reports has been minimal.

Early experiences with SP liver and pancreatic surgery indicate that the platform is technically feasible in selected cases, particularly for minor hepatectomies and distal pancreatectomies. Reported outcomes show low intraoperative blood loss, minimal conversion rates, and short hospital stays, with no significant increase in complications compared with multi-port robotic or laparoscopic surgery. However, the number of patients treated remains very small, and procedures are performed almost exclusively in high-volume centers by surgeons already experienced in multi-port robotics. Technical constraints—including limited triangulation, restricted stapler options, and difficulty with vascular control—currently limit application to peripheral or less complex resections. While these results are encouraging, the evidence remains preliminary, and long-term oncologic and functional outcomes are not yet available, preventing generalization to complex hepato-pancreatic procedures.

### 4.2. Cholecystectomy

Cholecystectomy is among the most common major surgical procedures performed worldwide. The advent of the da Vinci SP platform has rekindled interest in single-port robotic cholecystectomy (SPRC) for benign biliary disease with the potential for improved ergonomics, cosmetically appealing incisions, and reduced port-site morbidity. The recent literature has begun to establish technical outcomes, patient selection criteria, pain, operative time, and comparative performance with standard or multi-port techniques.

A large retrospective study including 304 patients (145 SP, 159 multi-port Xi) compared the SPRC with da Vinci SP vs. Xi systems for benign gallbladder disease. The SP group had mean operative times of 45.7 min and 49.8 min for multi-port Xi (*p* non-significant), with significantly shorter docking times for SP (5.7 min vs. 8.8 min; *p* = 0.024). Early postoperative pain scores were marginally lower in SP (median NRS 4.0 vs. 4.3; *p* = 0.003), though in the acute cholecystitis subgroup, pain was higher and operating time longer; morbidity and hospital stay were no different, however. These findings suggest that in benign disease, SP cholecystectomy is feasible and has perioperative outcomes comparable to or minimally better than multi-port procedures in well-selected patients [11].

A single-center series of 216 SPRC cases (compared to 118 with Si/Xi systems) also showed that SP allows for more complex disease (chronic or acute cholecystitis) without compromising safety. Total operative and docking times were considerably longer in SP, with no difference in postoperative complications, indicating that operative metric differences can favor SP when properly utilized [12].

The learning curve in SP cholecystectomy has been studied. CUSUM analysis in a recent study (SPRC) demonstrated that pre-console time, i.e., the setup before console use, and docking time reduce significantly over early cases, and overall operative time reduces from around 59 min in early phase to around 46 min in later phase (*p* < 0.001), with console time being relatively constant, showing transferability of console skills from other robotic or laparoscopic experience. Early outcomes (morbidity, conversion) did not worsen in early phases [13].

A propensity-matched retrospective multicenter study compared SP versus Xi system SIRC (single-incision robotic cholecystectomy), the older single-incision approach using da Vinci Xi. SP had shorter console time and pain scores in matched patients, while overall operative times and complications were similar. Estimated blood loss was somewhat greater in SP but with no clinically significant effects [13].

A comparative study of single-incision laparoscopic cholecystectomy (SILC) and single-incision robotic cholecystectomy (SIRC) using Xi and SP demonstrated that SILC had shorter operating times than robotic methods (~44.9 min vs. ~55 min), whereas robotic SP provided advantages on console time (lower with SP) and fully articulated arms, reducing surgeon effort. Complications and pain between groups did not differ significantly [14].

Side-by-side comparison with the prior Si robotic single-site systems showed the following differences: in 30 consecutive RSPC cases, mean dock time was ~5.2 min; dissection time ~14.6 min; operation time ~75.1 min; hospital stay ~1.5 days. These durations were significantly shorter than the Si single-site, which was ~109.5 ± 30.0 min for procedure, ~11.9 ± 4.3 min for docking, ~34.6 ± 18.4 min for dissection, and longer console time. Pain scores were also significantly better in RSPC [15].

While the SPRC with the SP system has promise, a number of challenges remain. Acute cholecystitis increases operative complexity, prolongs surgery, and intensifies early postoperative pain in such cases, although without necessarily affecting morbidity and hospital stay [12]. Careful selection of patients is therefore crucial, since most reported series include patients with only a modest BMI, limited prior surgery, or non-severe disease; obesity, dense adhesions, or severe inflammation could complicate SP procedures given their potential to affect instrument reach, visualization, or maintenance of pneumoperitoneum. Reports on pain and patient-reported outcomes beyond the early postoperative period are limited, as there are only modest benefits regarding incision-related pain or cosmetic results and weak support for functional recovery or long-term satisfaction. A number of technical considerations are also produced by the SP platform, such as the need for specialized instruments for insertion, docking, and sometimes additional ports in complex cases, although specimen extraction and control of bile spillage require careful planning due to limited accessory options. Lastly, the learning curve favors surgeons with prior experience in multi-port robotics or single-site laparoscopy, although early cases may be associated with longer docking, pre-console, and overall operative times, as would be expected, which improve with increasing experience [13].

Currently published experiences show that SP cholecystectomy has a high feasibility and safety, with conversion rates close to zero and postoperative outcomes no different from multi-port robotic and laparoscopic techniques. The progressive reduction in operative and docking time after the learning curve is a consistent finding, with some studies showing comparable or shorter docking times compared to the multi-port systems. Pain and cosmetic satisfaction tend to favor the single-port approach, although functional recovery benefits are still inconsistently reported. Collectively, these data indicate that SP cholecystectomy is a mature, reproducible procedure in well-selected patients, supported by growing evidence and a predictable learning curve [16,17,18,19,20,21].

### 4.3. Colorectal Resections

The aggregate clinical experience to date paints an overall positive picture of the da Vinci SP platform for colorectal surgery, albeit within the constraints of an evolving technology. In the first systematic review ever devoted to it—eleven studies and a total of approximately 396 patients were included—SP colorectal procedures were shown to be safe and feasible: the intra-operative complication rate was more or less 0.5%, and postoperative complications were encountered in 12% to 15%, one of the most frequent events being postoperative ileus [22].

The same findings were observed in a larger scoping review of 22 studies and approximately 465 patients, 82.6% of whom underwent surgery for malignancy. In the larger cohort, conversion to multi-port laparoscopy occurred in approximately 4.2% of procedures with no conversion to open surgery [23]. These figures are validated even in the “early adoption” phase by initial single-center experiences. For example, in a series of ten patients, the mean operative time was around 222 min, the hospital stay around six days, and postoperative complications were minimal [24].

All these reports, collectively, indicate that, in skilled hands and in properly selected patients, SP colorectal surgery is as safe as conventional laparoscopic or multi-port robotic techniques, with a comparatively low conversion rate.

Although few comparative studies are available, and these are marred by small sample sizes and retrospective designs, their results are nonetheless informative. In a case-matched comparison between single-port and multi-port robotic total mesorectal excision (TME) for rectal cancer, perioperative outcomes such as postoperative complication rates and pathologic quality of the specimen were essentially the same. Notably, the SP group also experienced slightly less intra-operative blood loss (around 20 mL versus 30 mL) and a marginally shorter hospital stay (seven versus eight days), although not all of the differences were statistically significant [25]. A single comparative study of left-sided colon cancer found that SP colectomy was associated with reduced intra-operative bleeding, faster return of bowel function, reduced postoperative analgesic need, and increased patient satisfaction with cosmetic outcomes when compared with the reduced-port “Single-Site” approach [26].

Though initially, these findings suggest that SP robotic surgery can achieve at least equivalent—if not slightly improved—short-term outcomes relative to both multi-port and other reduced-port methods, particularly with respect to postoperative comfort and cosmetic satisfaction.

From an oncologic standpoint, the data demonstrate that the single-port approach does not compromise the fundamental principles of radicality. In the scoping review by Picciariello et al., the mean lymph-node yield in cancer resections ranged from 15 to 28, a finding entirely within internationally accepted oncologic standards, and the R0 (margin-negative) resection rate exceeded 90% [23].

Similarly, the matched analysis by Jeong et al. showed no difference between SP and multi-port robotic TME for mesorectal quality or margin status of resection [25].

It must be mentioned, however, that most studies report less than twelve months of follow-up; thus, robust data on local recurrence, disease-free survival, overall survival, and long-term functional results are still lacking.

The current evidence across systematic reviews and comparative studies shows that SP colorectal surgery achieves perioperative outcomes comparable to multiport robotic and laparoscopic approaches, with low conversion rates (≈4–5%), acceptable complication profiles, and oncologic adequacy confirmed by R0 resection and appropriate lymph-node yield. A consistent pattern is longer operative time in the early learning phase, probably related to familiarization with the platform and availability of the stapler. Although short-term outcomes are encouraging and reproducible across institutions, long-term oncologic and functional results are still lacking, limiting generalizability beyond experienced centers.

### 4.4. Gastric Surgery

All clinical data are retrospective or at early stages. A prospective phase I/II single-arm trial (Park et al.) evaluated da Vinci SP-assisted minimal-port distal gastrectomy (SPRDG) in 19 patients with gastric cancer and attained successful completion with no severe complications or conversion; mean operation time was ~218 min, and mean stay was 3.2 days. The authors concluded SPRDG with the da Vinci SP was safe and feasible for selected patients with an experienced surgeon [27].

Single-center early experience series (excluding Japanese and East Asian centers in which minimally invasive gastrectomy is strongly established) validate these findings: early series establish procedural feasibility, low rates of major complications in selected populations, and no unwelcome signals of safety if conservatively selected patients and experienced teams are employed. Early clinical series with more recent publications also report uneventful total gastrectomy with SP platforms in carefully selected situations, demonstrating the expanding envelope of use beyond distal resections [28,29].

A broader scoping review and meta-analysis of SP use across general surgery (including gastrectomy) summarized current evidence up to 2024, noting SP-robotic procedures have been used safely across a range of operations but highlighting that current evidence is primarily observational and institutionally derived. The appraisal at present captures both the promise and current gaps in evidence for routine adoption in major abdominal oncologic resections [1].

For available series, single-port SP gastrectomy working times are variable—often longer in an early-experience period than multi-port laparoscopic or robotic methods—keeping in step with the learning curve and additional intracorporeal steps. Blood loss is generally minor and equivalent to a multi-port robotic gastrectomy. Hospitalization and recovery duration in early reports are encouraging (short hospital stay, rapid return of bowel function), but these are based on small, often highly selected groups and single-center pathways that may introduce biases (optimized recovery protocols, specialized teams). Comparative data are scarce; decreased-port robotic systems (using Xi or Single-Site techniques with extra ports) have shown less post-op pain but not consistently shorter operative time or improved oncologic results. Thus, while perioperative safety is to be commended in experienced hands, claims of superiority over mature multi-port techniques are premature [27,28,29,30,31].

Oncologic adequacy—as measured by margin status and lymph node count—is the most important requirement of any method used in gastric cancer surgery. Early SP series report adverse margins on resection specimens and lymph node procurement in ranges reported for multi-port robot-assisted gastrectomy in high-volume centers, but with short follow-up and small series. Reduced-port robotic series indicate that robot support can buffer the technical difficulty of lymphadenectomy and may preserve lymph node harvest compared to laparoscopy, but extrapolation of evidence to strict SP procedures must be approached with caution. Extended D2 dissection has defined technical difficulties (suprapancreatic and splenic hilum stations), and the majority of units still favor either reduced-port hybrid or multi-port robot technique for advanced, complex tumors. More extensive comparative studies with standardized pathology reporting and longer oncologic follow-up (recurrence, disease-free, and overall survival) are needed.

Current evidence demonstrates that SP gastrectomy is a technically feasible and safe procedure in well-selected patients, with low complications and short lengths of stay in the early series so far available. However, they all share small sample sizes, highly selected early gastric cancers, and were performed in high-volume centers with extensive prior robotic experience. Operative time tends to be longer at the beginning of the learning curve, reflecting the complexity of lymphadenectomy and the limits of SP-specific stapling instruments. Above all, while early oncologic surrogates, such as R0 margins and lymph-node yield, seem acceptable, long-term oncologic outcome data and functional data are still lacking, preventing generalization to broader gastric cancer surgery. The current status of SP gastrectomy should therefore be considered a promising but early-stage application awaiting broader validation.

## 5. Evidence Synthesis: Safety, Feasibility, and Perioperative Outcomes

Evidence-based outcomes are presented qualitatively rather than quantitatively due to heterogeneity in study design, outcome measures, and reporting standards. Table 1 consolidates key information to improve readability and allow comparison across target organs at a high level, with no implication of statistical comparability.

The literature for the da Vinci SP overall surgery is defined by small prospective case series, retrospective cohort comparisons, and registry reports by device-centricity; randomized controlled trials (RCTs) are scarce. Several systematic reviews in individual procedure categories (e.g., colorectal surgery, single-site cholecystectomy) concluded that the SP strategy is safe and feasible but reported that the quality of evidence is low to moderate and heterogeneous [18,22,32].

Initial experience generally demonstrates longer operative durations for SP procedures compared to well-known multi-port robotic or laparoscopic approaches in terms of the initial learning curve; with increased experience, docking and console times decrease and may equate to or even be shorter than comparator techniques for individual procedures [12,14,18,20]. Conversion rates to multi-port or open surgery tend to be low in published reports but are procedure and case-mix-dependent [11,24,26].

Learning curve studies, originally delineated in urologic series and in cholecystectomy series, suggest procedure-specific learning curves with earlier plateaus for simpler procedures (e.g., cholecystectomy) and longer acquisition curves for the more demanding colorectal operations. Institutional volume of cases and experience of the team (anesthetic team, scrub team, assistants) all influence the rate of proficiency significantly.

Studies in urology and cholecystectomy show a learning curve plateau after ~15–20 cases when surgeons have prior multi-port robotic experience, whereas complex colorectal cases may require >40 cases to reach proficiency. Structured proctoring significantly reduces console and docking times [33].

Published complication rates of SP cholecystectomy and colon resections are comparable to multi-port procedures in matched series, with no good evidence for greater major morbidity from the SP device [11,24,26]. Oncologic surrogates for oncologic surgery in colon resections (extent of resection, number of lymph nodes removed) have been within acceptable ranges in early series, but long-term follow-up for local recurrence and survival is not yet available [22,23].

Single-incision methods favorably report improved cosmetic outcomes and possible pain reductions with incisions; however, proof of clinically important pain reductions or earlier return of function is unclear and often confounded by analgesic regimens and patient preference [14,18,22,28]. Strong patient-reported results and systematic pain measurements need to be employed.

## 6. Instrumentation, Stapling, and Energy Devices: Supplementing the SP Armamentarium

Historically, a limitation of the SP platforms has been the lesser availability of vessel-sealing and energy devices, mission-critical to overall surgery. Regulatory approvals and device releases currently in place (such as single-port staplers and advanced energy devices designed for use with the SP) have meaningfully expanded the breadth of procedures feasible with the SP platform [1,2,3,4]. Device-specific stapler clearance within SP robotic systems has been emphasized by industry and clinical commentators as a key enabler for extended use across colorectal and thoracic procedures [1,2,3,4]. Being able to have such an adjunct on hand reduces hybrid conversion requirements and enhances procedural efficiency.

The SP platform maintains the ergonomic benefits of robotic surgery for the console surgeon and can potentially reduce outside clutter and port-site morbidity for patients. Internal triangulation from a single incision is a valuable technical advantage in cramped quarters like the pelvis or subdiaphragmatic recesses. Articulation of flexible scope offers multiple planes of view with no requirement for auxiliary ports. Some authors characterize augmented instrument reach to constrictive anatomy channels otherwise limited by external port shape in multi-port techniques [2,3]. These characteristics can be embodied as technical simplicity for some planes of dissection and allow natural orifice or single-site retrieval techniques.

## 7. Shortcomings and Limitations

Robotic platforms have major capital, maintenance, and per-case disposable costs. Historically, multi-port system cost analyses have determined high per-case consumable costs and fixed costs; adoption of SP has the added expense of device-specific instrumentation and training, and the cost–benefit ratio is not yet clearly favorable to all general surgery indications [13,34]. Hospitals must weigh anticipated reductions in length of stay or complications against incremental device amortization and consumables. Comparative cost-effectiveness data for SP against multi-port robotic and conventional laparoscopic methods, particularly in general surgery, are not yet available.

Despite articulation instruments, the SP system may have limitations in terms of instrument traction or force in extremely fibrotic or obese patients. Single fulcrum working can pose unique force vectors and may affect tissue handling in some cases. Surgeons will have to adapt to haptic variation and consider case selection appropriately.

Adoption is more than just individual surgeon training, as well as team skill in docking, instrument handover, bedside assistance, and problem-solving. Operating room logistics are different (single-site extraction techniques, retrieval of specimens), and institutions must invest in training courses and proctorship. Initial series indicate the requirement for proctored cases to decrease learning curves and avoidable conversions or complications [13,34].

Most SP series reported have an origin in high-volume centers with experienced robotic surgeons and are susceptible to selection bias towards healthier patients. Comparative effectiveness studies, randomized or registry-based, controlled for potential confounders such as BMI, prior surgery, and case severity, are few in number. Endpoint choice variation (e.g., cosmetic satisfaction, pain, hospital stay) also complicates synthesis.

Practice patterns currently suggest incremental implementation of SP general surgery, starting with lower complexity cases (e.g., elective cholecystectomy for uncomplicated disease) and advancing toward higher complexity colorectal resections as team experience and availability of instruments improve [13,33]. Patient factors to be considered before surgery are BMI, prior abdominal operations (adhesions), inflammatory status (e.g., acute cholecystitis increases technical difficulty), and complexity of disease.

Preoperative preparation should include imaging review for port placement, contingency docking if additional ports are needed, and clear conversion criteria. Proctoring of cases and training protocols should be included in institution-based credentialing tracks.

A major limitation of the current evidence is the absence of long-term follow-up. Across most studies, follow-up duration is <12 months, and data on cancer-specific survival, hernia recurrence, chronic postoperative pain, or quality of life are missing. Consequently, while SP seems safe in the short term, its long-term durability and oncologic equivalence remain unproven. Future trials should include standardized long-term outcomes (DFS/OS, recurrence, chronic pain, PROMs).

## 8. Future Directions and Priorities for Research

Table 2 summarizes all future directions and priorities for research.

While the da Vinci SP platform has become increasingly adopted for validated procedures as discussed earlier, several applications in general surgery remain preliminary or even theoretical, with no clinical outcomes having been published specific to SP:-*Ventral and complex incisional hernia repair*: While there are conceptual advantages of single-incision access, to date, no clinical SP series exist, and the available literature on ventral hernia refers to multiport robotic techniques and not SP [35,36,37]. Technical barriers, namely mesh handling and the need for large extraction or ancillary ports, currently remain limiting factors.-*Foregut surgery (fundoplication, paraesophageal hernia repair)*: Early small case series demonstrate feasibility and safety, although evidence is limited to short-term outcomes and highly selected patients [38,39]. Long-term results on reflux control or recurrence are not available.-*Splenectomy*: Early SP splenectomy experiences demonstrate that the procedure is technically feasible and safe in selected cases, particularly when spleen size is moderate and vascular control can be achieved through a single access. Reported outcomes show low intraoperative blood loss, short length of stay, and rapid postoperative recovery, reflecting the ergonomic benefit of single-port articulation in the left upper quadrant. However, the number of published cases remains extremely limited, and procedures are largely restricted to benign hematologic indications; cases of massive splenomegaly or trauma have not been evaluated. Current evidence, therefore, is preliminary and not generalizable due to small cohort size and lack of comparative data, pending larger prospective studies [40,41].-*Adrenalectomy*: SP adrenalectomy has shown high feasibility with very low conversion rates and excellent perioperative outcomes, particularly for benign adrenal lesions < 6 cm. The single-port approach seems to offer advantages, including less postoperative pain and improved cosmetic satisfaction, with comparable operative time and complication rates compared to multi-port robotic or laparoscopic adrenalectomy. The platform’s wristed instruments allow comfortable dissection in the confined retroperitoneal space, reducing the need for patient repositioning. However, all available studies are from small, highly selected cohorts and do not include complex tumors or malignancies requiring lymphadenectomies. Therefore, although SP adrenalectomy is emerging as a promising application, evidence is still insufficient to extend routine use beyond selected benign pathology [42,43].

These procedures are likely to expand with the improvement in SP-compatible instrumentation (e.g., mesh introducers, advanced staplers), structured training, and prospective studies with appropriate outcome reporting.

## 9. Discussion

This scoping review summarizes the current clinical evidence for the da Vinci^®^ Single-Port (SP) platform in general surgery. Throughout the abdominal procedures, the SP system consistently appeared to be feasible and safe when used in selected patients and by surgeons already experienced in minimally invasive surgery. In addition, short-term outcomes such as postoperative pain, length of stay, complication rates, and conversion showed comparable results with multi-port robotic or laparoscopic approaches. One recurrent finding has to do with the very low rate of conversions, which further speaks to the platform’s reliability even in the early stages of adoption.

However, the feasibility demonstrated in the literature should not be interpreted as universal applicability. The available evidence comes predominantly from small, single-center retrospective series, often involving highly selected cases. Operative time is frequently longer during the learning curve, especially in procedures requiring complex stapling angles or multi-quadrant access. On the other hand, when procedures are anatomically confined and benefit from a single incision, the SP platform apparently offers tangible advantages. For instance, in cholecystectomy and adrenalectomy, the SP system decreases port-related trauma, enhances ergonomics, and may improve cosmesis and postoperative pain. In colorectal and gastric surgery, while SP is technically feasible and oncologically adequate, the advantage over multi-port approaches is less clear and can be limited by instrument crowding, reduced triangulation, and stapling constraints.

These findings indicate that the added value of the SP approach is procedure-dependent. SP surgery seems to be most advantageous in cases where the surgical field is accessible through a single quadrant-including single-incision cholecystectomy and adrenalectomy in procedures where incision minimization has clinical or patient-centered relevance. On the other hand, for multi-quadrant procedures or complex oncologic resections, there is still uncertainty regarding the incremental benefit over standard robotic platforms, and data to support this are lacking.

While the included studies mention such adoption barriers as costs, training, and proctorship requirements, to date, there is no SP-specific cost-effectiveness evidence in general surgery. Available cost analyses pertain to multi-port systems or to the use of SP in urology and, therefore, cannot confidently be extrapolated to abdominal SP surgery. For this reason, the manuscript reports cost as a limitation rather than attempting speculative quantification.

Looking ahead, single-port robotic surgery is likely to benefit from rapid technological evolution and increasing competition. Intuitive Surgical has recently announced new instrumentation for SP, including stapling capability and expanded end-effectors, following FDA device clearance [3]. In parallel, other manufacturers are entering the single-access or reduced-port field: Titan Medical’s SP ‘Enos’ platform (Toronto, Ontario, Canada) was acquired by Medtronic in 2022 with the publicly stated goal of advancing single-access robotics, while Medtronic’s Hugo™ RAS (Dublin, Ireland)and CMR Surgical’s Versius^®^ (Cambridge, United Kingdom) have highlighted port-reduction strategies in their corporate development pipelines [4,44]. These publicly disclosed developments indicate that competition and instrument diversification may accelerate adoption, lower per-procedure costs, and expand the range of feasible SP procedures.

## 10. Conclusions

The da Vinci SP robotic system is an important evolution of minimally invasive surgery, allowing complex procedures to be undertaken through a single incision with articulating instruments and enhanced ergonomics. Based on current evidence, SP surgery is safe, reproducible, and clinically equivalent to multi-port robotic and laparoscopic techniques in the short term. The clearest benefits are seen in procedures inherently suited to a single incision, such as cholecystectomy and adrenalectomy, where reduced port trauma may translate into improved cosmesis, less pain, and faster recovery.

However, the current literature remains preliminary and underpowered. Most series include limited patient numbers, short follow-up periods, and selected cases performed by expert surgeons. Consequently, generalization to broader populations or complex oncologic surgery is premature. In addition, a lack of SP-specific cost-effectiveness studies limits conclusions about the platform’s value relative to standard robotic or laparoscopic systems.

Future research priorities include:Prospective registries and randomized trials comparing SP to multi-port robotics and laparoscopy;Procedure-specific clinical benefit evaluation beyond feasibility;Longer-term oncologic and functional outcomes;Assessment of training pathways and learning curve;Development of novel SP-dedicated instruments together with various single-port platforms.

In summary, SP robotic surgery is an innovative and promising modality that enjoys specific clinical strengths, although its optimal role in general surgery remains to be defined. Evidence up to now supports its use in selected indications, while further research is needed to confirm that the theoretical advantages will translate into meaningful clinical and economic benefit.

## Figures and Tables

**Figure 1 jcm-14-08212-f001:**
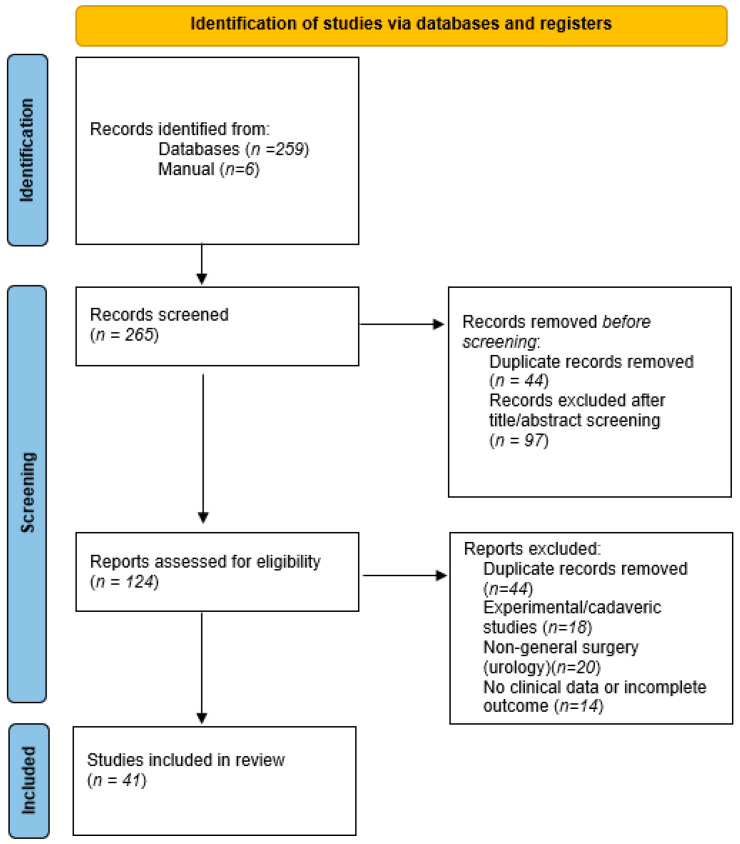
PRISMA-lite flowchart showing the selection of studies included in this scoping review.

**Table 1 jcm-14-08212-t001:** Qualitative synthesis of outcomes by target organ in SP robotic surgery.

Target Organ	Evidence Available	Key Qualitative (Non-Comparative) Findings
Liver and Pancreas	Case series. Early feasibility studies.	Low blood loss. Low conversion. Feasibility is mainly in small lesions.
Gallbladder	Largest experience. Multiple retrospective studies.	Comparable safety to multiport. Reports of reduced pain and improved cosmesis.
Colon and Rectum	Early feasibility. Mainly single-center cohorts.	Safe. Oncological adequacy preserved. Longer learning curve.
Stomach	Feasibility studies. Limited number of cases	Longer operative time during the learning curve. Feasible in early gastric cancer.
Adrenal gland	Feasibility reports	Low morbidity. Single access is particularly advantageous in confined anatomy.
Other procedures (Hernia, Splenectomy, Foregut)	Case reports	Too heterogeneous to assess outcomes. Preliminary experience.

**Table 2 jcm-14-08212-t002:** Future directions and priorities for research.

Registries and randomized controlled trials	Standardized variable multi-center registries and procedure-specific RCTs (SP vs. multi-port robotic and conventional laparoscopic) would increase real-world evidence generation in addition to strong comparative effectiveness and safety data.
Standardized outcomes	Use of standardized outcomes (e.g., Clavien–Dindo for complications, validated patient-reported outcomes for pain and cosmesis, quality of life metrics) will strengthen evidence synthesis.
Cost-effectiveness analysis	In-depth economic modeling for institutional volume, device amortization, and downstream cost savings (fewer complications, faster recovery) will establish value propositions.
Instrumentation and adjunct development	Continued evolution of SP-compatible staplers, suturing devices, and energy platforms will expand the list of potential procedures.
Long-term results	Incisional hernia rates, wound complications, oncologic recurrence, and long-term functional outcomes will be essential.
Training science	Comparative training modality and credentialing framework study for SP platforms will facilitate safer dissemination.

## Data Availability

No data were created.

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
