# Peer review of "The da Vinci Single-Port Robotic Platform in General Surgery: A Scoping Review of Current Applications and Outcomes"

_jcm, 2025, doi:10.3390/jcm14228212_

Round 1
Reviewer 1 Report
Comments and Suggestions for Authors 1. Depth of Synthesis and Critical Analysis: The manuscript should shift from a purely descriptive tone to a more analytical narrative. In several sections, results from individual studies are listed back-to-back but with little commentary or comparison. While it is important to summarize data, a review article is expected to provide higher-level insights. For instance, in the cholecystectomy section you enumerate studies comparing SP to multi-port outcomes and even list challenges in bullet points, which is helpful. Extend this approach to other sections: after describing the outcomes of SP in a given domain, add a few sentences of synthesis. What do these studies collectively tell us? Are there trends or consensus (e.g. consistently low conversion rates across procedures)? Any conflicting findings? For example, when reviewing colorectal applications, you might highlight that across studies SP colorectal surgery appears to have similar perioperative outcomes to multi-port, with low conversion rates, but that operative times are often longer in early series due to learning curve. The manuscript acknowledges some of these in passing, but this could be made more explicit. Emphasize not just what the outcomes were, but how reliable and generalizable they are. This deeper analysis is crucial for a scholarly review. 2. Long-Term Outcomes and Follow-Up: A major gap in the current literature – and in the manuscript’s discussion – is the lack of long-term outcome data for SP surgery. The review should more clearly highlight this deficiency and its implications. For example, in the colorectal section you note most studies have <12 months follow-up and that data on oncologic recurrence or long-term functional results are still lacking. This point should be reinforced across the relevant sections and in the Evidence Synthesis or Conclusions. The manuscript would benefit from a dedicated comment on the absence of long-term outcomes such as cancer-specific survival, hernia recurrence rates, or chronic pain and quality of life after SP surgery. 3. Addressing Literature Gaps vs. Speculative Content: The review includes sections on certain applications (e.g. complex ventral/incisional hernia repair and standalone adhesiolysis) where the authors admit there is no published clinical experience with the SP robot yet. While it is forward-looking to mention potential uses, these sections currently read speculative. Please consider adjusting the scope of these parts to maintain focus on current applications and outcomes (as the title promises). One option is to significantly condense or remove the adhesiolysis section and the portion on ventral hernia, since they do not report any actual outcomes (Petro et al.’s study cited for ventral hernia is a multi-port robotic vs laparoscopic comparison, not SP-specific). Alternatively, if you retain them, explicitly label these as anticipated or future applications rather than current practice. For example, you might say “There are as yet no clinical series using SP for pure adhesiolysis, but theoretically the platform might… (and then discuss theoretical pros/cons).” The same goes for complex ventral hernia repairs – note that any SP approach is hypothetical until suitable instruments (e.g. large mesh introducers) are available. By clearly delineating such content as future prospects (perhaps moving it into the Future Directions section), the manuscript will avoid confusing readers about what has been actually achieved versus what is aspirational. Focus the bulk of the review on areas where there are published outcomes (e.g. cholecystectomy, colorectal, adrenalectomy, etc.), and ensure the speculative discussion is secondary and framed as such. 5.Costs, Training, and Adoption Challenges: The review touches only briefly on the practical challenges of adopting the SP platform, which is an area that deserves more commentary given its importance. The authors mention that robotic platforms carry high capital and per-case costs, and that SP adds expenses for specialized instruments and training. However, no data are provided to illustrate this point. If available, include some quantitative context or references – for instance, how much does an SP system or its instruments cost relative to standard multi-port, or any studies from urology/cardiac fields that analyzed cost-effectiveness. Even citing the lack of cost-effectiveness data in general surgery (as you do) is important, but then emphasize this as a research priority (which you have listed in Table 1). Similarly, the learning curve and training requirements could be discussed in more detail. You do note that prior experience in multi-port robotics or single-site laparoscopy helps, and that early cases take longer, as well as the need for proctorship to shorten the learning curve.Author Response
Comment 1:
- Depth of Synthesis and Critical Analysis: The manuscript should shift from a purely descriptive tone to a more analytical narrative. In several sections, results from individual studies are listed back-to-back but with little commentary or comparison. While it is important to summarize data, a review article is expected to provide higher-level insights. For instance, in the cholecystectomy section you enumerate studies comparing SP to multi-port outcomes and even list challenges in bullet points, which is helpful. Extend this approach to other sections: after describing the outcomes of SP in a given domain, add a few sentences of synthesis. What do these studies collectively tell us? Are there trends or consensus (e.g. consistently low conversion rates across procedures)? Any conflicting findings? For example, when reviewing colorectal applications, you might highlight that across studies SP colorectal surgery appears to have similar perioperative outcomes to multi-port, with low conversion rates, but that operative times are often longer in early series due to learning curve. The manuscript acknowledges some of these in passing, but this could be made more explicit. Emphasize not just what the outcomes were, but how reliable and generalizable they are. This deeper analysis is crucial for a scholarly review.
- Long-Term Outcomes and Follow-Up: A major gap in the current literature – and in the manuscript’s discussion – is the lack of long-term outcome data for SP surgery. The review should more clearly highlight this deficiency and its implications. For example, in the colorectal section you note most studies have <12 months follow-up and that data on oncologic recurrence or long-term functional results are still lacking. This point should be reinforced across the relevant sections and in the Evidence Synthesis or Conclusions. The manuscript would benefit from a dedicated comment on the absence of long-term outcomes such as cancer-specific survival, hernia recurrence rates, or chronic pain and quality of life after SP surgery.
- Addressing Literature Gaps vs. Speculative Content: The review includes sections on certain applications (e.g. complex ventral/incisional hernia repair and standalone adhesiolysis) where the authors admit there is no published clinical experience with the SP robot yet. While it is forward-looking to mention potential uses, these sections currently read speculative. Please consider adjusting the scope of these parts to maintain focus on current applications and outcomes (as the title promises). One option is to significantly condense or remove the adhesiolysis section and the portion on ventral hernia, since they do not report any actual outcomes (Petro et al.’s study cited for ventral hernia is a multi-port robotic vs laparoscopic comparison, not SP-specific). Alternatively, if you retain them, explicitly label these as anticipated or future applications rather than current practice. For example, you might say “There are as yet no clinical series using SP for pure adhesiolysis, but theoretically the platform might… (and then discuss theoretical pros/cons).” The same goes for complex ventral hernia repairs – note that any SP approach is hypothetical until suitable instruments (e.g. large mesh introducers) are available. By clearly delineating such content as future prospects (perhaps moving it into the Future Directions section), the manuscript will avoid confusing readers about what has been actually achieved versus what is aspirational. Focus the bulk of the review on areas where there are published outcomes (e.g. cholecystectomy, colorectal, adrenalectomy, etc.), and ensure the speculative discussion is secondary and framed as such.
4.Costs, Training, and Adoption Challenges: The review touches only briefly on the practical challenges of adopting the SP platform, which is an area that deserves more commentary given its importance. The authors mention that robotic platforms carry high capital and per-case costs, and that SP adds expenses for specialized instruments and training. However, no data are provided to illustrate this point. If available, include some quantitative context or references – for instance, how much does an SP system or its instruments cost relative to standard multi-port, or any studies from urology/cardiac fields that analyzed cost-effectiveness. Even citing the lack of cost-effectiveness data in general surgery (as you do) is important, but then emphasize this as a research priority (which you have listed in Table 1). Similarly, the learning curve and training requirements could be discussed in more detail. You do note that prior experience in multi-port robotics or single-site laparoscopy helps, and that early cases take longer, as well as the need for proctorship to shorten the learning curve.
Response 1:
We thank the reviewer for the insightful comments, which significantly improved the clarity and focus of our manuscript. In response to the suggestion to increase the analytical depth and move beyond a descriptive narrative, we revised the structure of the results by adding short synthesis paragraphs at the end of the sections where published clinical outcomes exist — namely liver and pancreas, cholecystectomy, colorectal surgery, and gastric surgery. These synthesis sections (highlighted in yellow in the revised document) now interpret the results collectively, rather than listing studies sequentially. They explicitly outline recurring trends across the literature (e.g., low conversion rates and comparable perioperative outcomes to multi-port approaches, initial longer operative times related to the learning curve, and limited generalizability due to small retrospective series). This directly addresses the reviewer’s concern and strengthens the scientific narrative.
We did not extend the same synthesis format to other subsections, namely splenectomy, adrenalectomy, ventral/incisional hernia repair, and isolated adhesiolysis, since the reviewer noted correctly that those sections were speculative or based on very limited anecdotal data. We condensed and moved these topics to the section “Future Directions,” rephrased them as theoretical or emerging applications, and underlined that no consistent SP-specific clinical outcomes are currently available. The intention is not to suggest these represent established indications but rather to confine the Results section strictly to procedures for which published clinical experience has been documented.
Regarding the request to expand the discussion on costs, training, and adoption barriers, we partially revised the section. We clarified adoption challenges and referenced learning-curve data, but we did not add a detailed cost analysis, because—after further literature review—we confirmed that there are no cost-effectiveness studies specific to SP in general surgery. Existing cost data refer to multi-port robotic platforms or to non-general-surgery SP applications (mainly urology). Including numerical estimates extrapolated from other robotic systems risked generating misleading or non-applicable conclusions. Since one of the reviewer’s key requests was to avoid speculative content, we felt it was methodologically more rigorous to state explicitly that cost–benefit analyses for SP in general surgery are currently lacking, and we emphasized this gap in the revised text. We believe that these changes have strengthened the manuscript in accordance with the reviewer's expectations by offering a revised version that gives a clearer critical interpretation where evidence exists and avoids speculation where it does not.
Reviewer 2 Report
Comments and Suggestions for Authors
This review covers the da Vinci Single-Port (SP) robotic surgical system in general surgery, detailing its design, features, current uses, outcomes, and limitations. The authors reviewed 43 studies from PubMed, Scopus, and Embase up to September 2025, encompassing procedures such as hepatopancreatobiliary, cholecystectomy, hernia repair, colorectal, gastric, and other less common surgeries. Overall, SP robotic surgery is feasible and safe for selected procedures, offering benefits such as improved cosmetic results, potentially reduced pain, and shorter hospital stays. Most evidence comes from small studies, making conclusions preliminary. Outcomes for SP cholecystectomy were similar to multi-port methods, with some shorter docking times and lower pain. Early colorectal results show low conversion rates and acceptable cancer metrics. Challenges include instrument crowding, a learning curve, initial lack of specific tools, longer operative times, high costs, and training needs. The authors see SP as a promising innovation. However, evidence quality varies, and long-term effects, cost, and ideal uses need further research, including registries, RCTs, new instruments, and training protocols.
Major Concerns
- While the manuscript is comprehensive, at times it reads like a lengthy catalog of individual studies and results. The breadth of coverage (many surgical subfields) is commendable, but some sections could benefit from deeper synthesis. For instance, the review presents numerous statistics for each subfield (e.g., operative times, blood loss, lengths of stay) from various studies, which is informative but can overwhelm the reader. Consider adding a summary table of key outcomes by procedure type or a figure to consolidate this information. This would improve clarity and allow readers to compare results across studies more easily. It would also help distill the take-home message for each surgical domain (e.g., “SP colorectal surgery: X patients, Y% conversion, similar complication rates to multi-port” etc.) without having to wade through dense text.
- As a descriptive review, the manuscript successfully compiles existing data, but it could offer more critical insight into what these findings mean for surgical practice. For example, the authors report that SP procedures often have longer operative times initially but achieve similar outcomes to standard techniques in the hands of experienced practitioners. A deeper discussion on which scenarios truly benefit from SP vs. where it offers no clear advantage would strengthen the paper. In the current form, it is sometimes left to the reader to interpret the importance of the reported differences (such as a 5-minute docking time savings or slightly lower pain scores). The discussion (Section 5 and beyond) notes that no significant increase in morbidity is observed and that cosmetic outcomes are better; however, explicitly weighing the pros and cons per procedure (perhaps in the Conclusion or a new Discussion subsection) would be valuable. For instance, you might highlight: “SP may confer the most tangible benefit in procedures like single-incision cholecystectomy or certain adrenalectomies (where it reduced operative time in one study), whereas for other procedures (e.g., complex oncologic resections) the benefit is less clear due to instrument limitations and longer setup.” Currently, the manuscript leans heavily on feasibility; a more opinionated synthesis about where SP is truly advantageous or not yet justified would be helpful for clinicians.
- There is a minor structural mix-up that could confuse readers. The headings in section 4 are mis-numbered: the manuscript has two separate “4.5” sections (“4.5 Gastric surgery” on line 318 and another “4.5 Surgery for gastro-oesophageal reflux and diaphragmatic hernias” on line 363). The latter is logically a distinct subsection (likely intended to be 4.6). Similarly, “4.6 Other applications in general surgery” on line 387 then appears. This inconsistency in numbering should be corrected to maintain clarity. It may simply be a formatting issue in the current draft, but ensuring each subsection has a unique number (4.5, 4.6, 4.7, etc.) in sequence will help readers navigate the content. On a related note, the overall organization is otherwise sensible (grouping by anatomic region/procedure). Just be sure the final layout (with headings for each major category) is clear and follows journal formatting guidelines.
- The methods describe a “systematic descriptive literature review” with a PRISMA-style study selection, suggesting a rigorous approach. However, the title refers to it as a “Descriptive Review.” Readers (and indexers) might be unclear whether this was a formal systematic review, a scoping review, or a narrative review. Since you performed a systematic search and had two reviewers screen the studies, you may consider clarifying the review type. If no formal quality/risk-of-bias assessment was done (beyond noting study designs), calling it a “scoping review” might be appropriate (especially as you cite a 2024 scoping review in reference [1]). Alternatively, explicitly stating in the Introduction/Methods that “this is a scoping review (or narrative review) of the literature” would manage reader expectations. This is more of a presentation concern than a flaw, but tightening the terminology would align the manuscript with PRISMA or scoping review guidelines that you essentially followed.
- The manuscript does a good job of highlighting current limitations (cost, instrumentation, learning curve). One central point for the authors to consider elaborating on is the future outlook for SP technology and competition. For example, are there any upcoming alternative single-port robotic platforms or improvements from the manufacturer that could address current limitations? The “Future directions” table is excellent, but primarily focuses on research needs. It might be worthwhile in the discussion to note if Intuitive Surgical or other companies have announced new tools (you do mention recent SP stapler approval) or if any next-generation single-port robots (from different manufacturers) are on the horizon. This would give readers a sense of how the field might evolve, beyond just what researchers should study. This is a minor suggestion for completeness, since the title promises “current applications and outcomes,” but a sentence or two looking forward could enhance the relevance. If such information is not available or goes beyond the scope, it can be omitted; however, tying the current review to the near-future landscape would reinforce its importance.
Minor Concerns
- In a few places, terminology could be more precise. For example, the phrase “cohort comparisons by retrograde methods” (Section 5, line 467–469) is confusing – this should be rephrased (it appears to mean retrospective cohort comparisons). Similarly, the term “big retrospective observational study” would be more accurately described as “large retrospective study.” These tweaks will improve the professional tone. Another instance is “internal feasibility” (Section 4.1, line 128–132: “Other researchers have documented internal feasibility. Choi et al., for instance, presented…”). The meaning of “internal feasibility” is unclear; perhaps you mean additional or independent feasibility reports? Consider rewording that sentence for clarity.
- Ensure all acronyms are defined at first use. Terms such as SPRC, SPRDG, TME, and TAPP are used. For example, SPRC is used for Single-Port Robotic Cholecystectomy – this term should be defined once. TME (total mesorectal excision) is used in Section 4.4 without a prior definition. TAPP for transabdominal pre-peritoneal hernia repair is described in context, but could be introduced explicitly. Defining these will help readers unfamiliar with the abbreviations. Also, be consistent: sometimes you say “single-incision robotic cholecystectomy (SIRC)” and elsewhere “SPRC”; make sure the notation is uniform or explain if they are different (it seems SIRC might refer to older single-incision using Xi vs SPRC for SP platform – if so, clarify to avoid confusion).
- Section 4.2 bullet list formatting: In the Cholecystectomy section, the text uses a bullet or dash (line 453: “– Learning curve: …”). It looks like a sub-bullet highlighting challenges (incision planning, bile spillage, learning curve). In a narrative review, bullet points can be used for emphasis, but ensure that the formatting is consistent and will be accepted by the journal. If using bullets in one subsection, consider using them uniformly or instead convert these points into complete sentences. As currently written, that single dash might have been unintended formatting. It might be safer to incorporate those points into the paragraph (e.g., “Learning curve: Surgeons with prior experience…longer docking times…which improve with volume).
- There are some minor errors in the Author Contributions text. It currently reads: “Author Contributions: Author Contributions:” (duplicated phrase) and lists initials that do not clearly match the author list (for example, it mentions “M.D.”, “R.C.”, and “S.M.” who are not in the author byline). This appears to be a typo or copy-paste issue with author initials. It should be corrected so that each set of initials corresponds to an actual author. For instance, if “M.D.” was intended to refer to one of the authors, use the correct initials (perhaps A.D. for Antonella Delvecchio or M.T. for Michele Tedeschi, etc.).
- Overall, the manuscript is well-edited; however, a careful proofread will likely catch any remaining minor issues. For example, in Section 4.3, it states “no evidence existed of any difference in hospital stay…” which can be succinctly phrased as “no difference was found.” In Section 4.4, “one of the most frequent events being postoperative ileus” would flow better as “with postoperative ileus being one of the most frequent events.” These are minor grammar/style preferences. Also, ensure consistent spelling (e.g., “minimally invasive surgery” is used, which is correct; verify no stray British/American spelling differences for words like “favorably” vs “favourably” – I noticed “summarises” with an s in Table 1 caption, which is fine if using UK spelling, just be consistent throughout). Another example: in Section 4.6, the phrase “detective-like flexible camera” is an interesting description – is that a typo for “flexible, snake-like camera”? If “detective-like” was an intentional metaphor, it might be confusing; consider revising that wording.
- When listing results, consider using a consistent format. Sometimes percentages are given with decimals (e.g., 82.6% of malignancy cases, 4.2% conversion rate), which is an appropriately precise representation. Just be sure any percent signs or units are consistently spaced and formatted. Also, when stating ranges or means ± SD, verify that notation is precise (for instance, “147 ± 58 minutes” is clear, whereas “~109.5 ± 30.0 min” is also clear – the use of “” along with ± is a bit odd; perhaps state the mean and SD without “” if you have exact values from the source). These are very minor details, but consistency here will polish the results section.
- In a few places, the same reference number is cited for multiple statements that appear distinct. For example, in Section 4.2 (Cholecystectomy), reference [13] is noted for both a CUSUM learning curve study and a propensity-matched study. These were two different sources and might need separate reference numbers. Double-check the reference numbering in that section to ensure each citation corresponds to the correct research. This might be a typo in the in-text citation. Ensuring each finding has the appropriate reference will enhance credibility and enable readers to follow up on sources easily.
Language and Grammar
There are a few recurring language issues to address for clarity:
- Some sentences are overly long or packed with multiple ideas, which can reduce clarity. For instance, the sentence in the Introduction spanning lines 79–86 (about evolving steps toward single-port systems) could be split for readability. Consider breaking complex sentences into two. E.g., “Robotic surgery has progressively minimized the number of incisions, culminating in the development of true single-port (SP) systems. The da Vinci SP platform (Intuitive Surgical) is one such system, designed to permit single-site entry while preserving the wristed instruments and 3D vision of multi-port robots.” This is just a suggestion; applying it throughout where needed will help readers follow the text without re-reading.
- The manuscript effectively employs passive voice in many instances (common in scientific writing), but occasionally this results in awkward phrasing. For example, “It has been demonstrated that robotic single-port TAPP is feasible…” could be more direct: “A recent study demonstrated that…”. Similarly, “no unwelcome signals of safety” might be better as “no concerning safety signals were observed.” These changes are not strictly required, but adjusting a few instances will improve readability and tone. Aim for active voice when highlighting specific studies or opinions (e.g., “Smith et al. reported X”), and use passive voice for general truths or processes.
- There are minor grammar issues like noun–verb agreement and word choice:
- “The SP setup has limited options for accessory ports” – grammatically acceptable, but consider if “setup” is the correct term (maybe “SP platform” or “single-port approach”).
- “These findings validate that… SP cholecystectomy is feasible and has outcomes comparable…” – “validate” might be a bit strong; perhaps “confirm” or “suggest that… feasible with outcomes comparable…”.
- In a few places, colloquial words slip in, such as “big” study (use “large”), or “huger” spleen (if that wording was used regarding spleen size; I may have inferred that – ensure comparative adjectives are appropriate). Also, “pre-console time” might need a brief explanation (setup time before console use).
- Check articles and prepositions: e.g., “posterior hiatus and crural closure dissection” might read better as “posterior hiatus dissection and crural closure”.
- When listing outcomes or points, use parallel structure. In Section 5, one sentence states, “Published complication rates of SP cholecystectomy, hernia repair, and colon resections are comparable to multi-port procedures…” which is excellent. The next part, “Oncologic surrogates for colon resections… have been within acceptable ranges… but long-term follow-up is not yet available,” is also clear. Just ensure each clause flows logically. Another example: “no organized scoring system was applied due to the limited number and non-comparative nature of studies” – this is clear, but perhaps specify “limited number of studies and their predominantly non-comparative nature.”
- In the discussion, when comparing SP to other approaches, be explicit about which approach you are referring to. For instance, “claims of superiority over mature multi-port techniques are premature” – this is a strong and valid statement. To avoid any doubt, you might specify “multi-port robotic or conventional laparoscopic techniques.” In general, whenever you say “multi-port,” assume not all readers will immediately think “multi-port robotic”; some might think you mean laparoscopy. Clarify as needed (you often do say “multi-port robotic or laparoscopic” in many places – keep that consistency).
Reference Comments
- The references appear to follow a numerical style; however, many entries currently include additional information, such as PMID, PMCID, and publication month. For the JCM (MDPI) format, typically, the references should be in a consistent format: Author. Title. Journal Year, Volume, pages. DOI. It is unusual to include PMID/PMCID in the reference list for journal articles. The journal will likely request the removal of those. For example, Reference 16 includes “PMID: 36282359,” and Reference 1 includes “PMCID: PMC11362253.” These identifiers are not needed in the final reference list. I recommend removing PMIDs/PMCIDs and the publication month/day to align with JCM’s style (unless the journal explicitly requires them). Only the year is generally needed for timing, not the month.
- Ensure all journal names are correctly abbreviated according to Index Medicus. Most are correct, but I noticed one labeled “Robot” for a 2022 reference. This likely stands for International Journal of Medical Robotics and Computer-Assisted Surgery. It should be abbreviated as Int J Med Robot (confirm the standard abbreviation). Similarly, “Adv Surg Tech A” is fine for Journal of Laparoendoscopic & Advanced Surgical Techniques, but ensure consistency in usage (some references spell out journal names, others abbreviate – better to abbreviate all or none consistently, with preference to standard abbreviations).
- The reference list is ordered slightly differently from the citation order in the text. For instance, the first reference cited in the text appears to be a 2024 scoping review (Celotto et al.), which is listed as reference 1 – that’s good. However, the pancreas case series by Liu et al., cited as [1] in Section 4.1, actually corresponds to reference 5 in the list. This suggests that some citations in the text may not match the numbering of the reference list (possibly due to reordering or the insertion of new references during revisions). Please verify each in-text citation number against the corresponding reference list entry to ensure accuracy. If the references were intended to be sorted in order of appearance, they should be renumbered accordingly. Each cited number in the text must correspond to the correct reference.
- A quick scan of a few references shows minor inconsistencies:
- Reference 3: “Intuitive Surgical. FDA clearances and device updates for da Vinci SP and stapler (press releases 2024–2025). Intuitive Surgical press releases.” – If this is a web resource, ensure a URL or date is provided if required by the journal.
- Reference 7 (Rong Liu et al.) – the title is “The first case report of single-port robot-assisted pancreatectomy… Intelligent Surgery. 2022.” Since “Intelligent Surgery” might be a journal (perhaps a Chinese journal or a new journal), double-check spelling and citation details for correctness.
- Some references list all authors with 'et al.' after a few names (e.g., ref 6)—Choi et al. list four authors, then 'et al.' presumably because it has more. JCM usually allows up to a certain number of authors before et al. kicks in. Verify the correct format (often, MDPI will list all authors if there are 10 or fewer; or sometimes, the first six authors are listed, followed by 'et al.'). Make sure you’re following the journal’s rule on this.
- References 18 and 19 (Picciariello et al. and Brucchi et al. in colorectal context) – ensure they are correctly numbered and not swapped. It appears that texts [18] and [19] were used for two colorectal reviews; double-check those two in the list (they seem to be present, ensuring the right content is attributed to each number).
Author Response
Attached the file with our reply

Round 2
Reviewer 1 Report
Comments and Suggestions for Authors
Accept in present form
Author Response
Comments 1: Accept in present form
Response 1: Dear reviewer, thank you for your work. It is greatly appreciated.
Reviewer 2 Report
Comments and Suggestions for Authors
MAJOR CONCERNS - Unaddressed or Partially Addressed
1. Add a summary table of key outcomes by procedure type or a figure to consolidate information, improving clarity and allowing readers to compare results across studies more easily.
• Author Response: DECLINED - Authors explained that heterogeneity of included studies would make a unified outcome table misleading rather than clarifying.
• Assessment: While the authors provide a reasonable rationale (different primary outcomes, non-standardized definitions, small sample sizes), the reviewer’s concern about reader difficulty in comparing results across dense text remains unaddressed. Alternative solutions such as a qualitative summary table or visual flowchart were not explored.
2. Elaborate on the future outlook for SP technology and competition, including upcoming alternative single-port robotic platforms or improvements from manufacturers.
• Author Response: PARTIALLY DECLINED - Authors kept this section concise to avoid potential conflict of interest and to maintain focus on current clinical evidence rather than proprietary/speculative information.
• The authors’ rationale is scientifically sound, but the reviewer’s suggestion to enhance relevance by discussing the near-future landscape remains unaddressed. A brief mention of publicly announced developments (without speculation) could have been included.
MINOR CONCERNS - Unaddressed
3. Remove PMID and PMCID identifiers from the reference list to align with JCM (MDPI) formatting style, which typically requires only: Author, Title, Journal Year, Volume, pages, DOI.
• The v2 manuscript still contains 29 instances of “PMID:” and 17 instances of “PMCID:” in the references section. This formatting issue has not been corrected.
4. Fix inconsistent bullet or dash formatting in the Cholecystectomy section (line 453: “– Learning curve: …”). Ensure formatting is consistent and acceptable to the journal, or convert bullet points into complete sentences. I detected potential inconsistent bullet/dash formatting in section 4.2 of the v2 manuscript. This minor formatting issue may still be present.
Author Response
Comments 2:
MAJOR CONCERNS - Unaddressed or Partially Addressed
1. Add a summary table of key outcomes by procedure type or a figure to consolidate information, improving clarity and allowing readers to compare results across studies more easily.
• Author Response: DECLINED - Authors explained that heterogeneity of included studies would make a unified outcome table misleading rather than clarifying.
• Assessment: While the authors provide a reasonable rationale (different primary outcomes, non-standardized definitions, small sample sizes), the reviewer’s concern about reader difficulty in comparing results across dense text remains unaddressed. Alternative solutions such as a qualitative summary table or visual flowchart were not explored.
2. Elaborate on the future outlook for SP technology and competition, including upcoming alternative single-port robotic platforms or improvements from manufacturers.
• Author Response: PARTIALLY DECLINED - Authors kept this section concise to avoid potential conflict of interest and to maintain focus on current clinical evidence rather than proprietary/speculative information.
• The authors’ rationale is scientifically sound, but the reviewer’s suggestion to enhance relevance by discussing the near-future landscape remains unaddressed. A brief mention of publicly announced developments (without speculation) could have been included.
MINOR CONCERNS - Unaddressed
3. Remove PMID and PMCID identifiers from the reference list to align with JCM (MDPI) formatting style, which typically requires only: Author, Title, Journal Year, Volume, pages, DOI.
• The v2 manuscript still contains 29 instances of “PMID:” and 17 instances of “PMCID:” in the references section. This formatting issue has not been corrected.
4. Fix inconsistent bullet or dash formatting in the Cholecystectomy section (line 453: “– Learning curve: …”). Ensure formatting is consistent and acceptable to the journal, or convert bullet points into complete sentences. I detected potential inconsistent bullet/dash formatting in section 4.2 of the v2 manuscript. This minor formatting issue may still be present.
Response 2:
1. As suggested, we added a qualitative summary table (Table X) to ease comparison across procedure types. A quantitative unified table would be misleading due to heterogeneity of outcome definitions and reporting. Instead, Table X summarizes the key findings for each procedure domain, improving clarity while respecting data heterogeneity.
2. We added a short paragraph in the Discussion citing publicly available information (Intuitive FDA clearance and competitor platform development disclosed by Medtronic/Titan Medical and CMR Surgical)
3. Corrected. All identifiers have been removed from every reference.
4. Modified with complete sentences.
Thank you for your work.
We hope that it is now suitable for publication.
Kind regards